# Branched chain amino acids harbor distinct and often opposing effects on health and disease

Christy L. Avery [1,2✉], Annie Green Howard [2,3], Harold H. Lee[1], Carolina G. Downie[1], Moa P. Lee[1], Sarah H. Koenigsberg [1], Anna F. Ballou [1], Michael H. Preuss [4], Laura M. Raffield [5], Rina A. Yarosh[1], Kari E. North [1], Penny Gordon-Larsen [2,6] & Mariaelisa Graff [1]

## Abstract

**Background** The branched chain amino acids (BCAA) leucine, isoleucine, and valine are essential nutrients that have been associated with diabetes, cancers, and cardiovascular diseases. Observational studies suggest that BCAAs exert homogeneous phenotypic effects, but these findings are inconsistent with results from experimental human and animal studies.

**Methods** Hypothesizing that inconsistencies between observational and experimental BCAA studies reflect bias from shared lifestyle and genetic factors in observational studies, we used data from the UK Biobank and applied multivariable Mendelian randomization causal inference methods designed to address these biases.

**Results** In $n = 97,469$ participants of European ancestry (mean age = 56.7 years; 54.1% female), we estimate distinct and often opposing total causal effects for each BCAA. For example, of the 117 phenotypes with evidence of a statistically significant total causal effect for at least one BCAA, almost half (44%, $n = 52$) are associated with only one BCAA. These 52 associations include total causal effects of valine on diabetic eye disease [odds ratio = 1.51, 95% confidence interval (CI) = 1.31, 1.76], valine on albuminuria (odds ratio = 1.14, 95% CI = 1.08, 1.20), and isoleucine on angina (odds ratio = 1.17, 95% CI = 1.31, 1.76).

**Conclusions** Our results suggest that the observational literature provides a flawed picture of BCAA phenotypic effects that is inconsistent with experimental studies and could mislead efforts developing novel therapeutics. More broadly, these findings motivate the development and application of causal inference approaches that enable 'omics studies conducted in observational settings to account for the biasing effects of shared genetic and lifestyle factors.

## Plain language summary

The three branched chain amino acids (BCAAs) leucine, isoleucine, and valine are important building blocks of muscle proteins that are obtained from the diet. Many studies in human populations have examined whether BCAAs affect health and disease. These human studies report results that are inconsistent with results from highly controlled animal studies. Because interest in the therapeutic targeting of BCAAs is growing, we wanted to better understand these discrepancies. Briefly, we used data from a large database that captured many diseases (e.g., cardiovascular disease, cancers, and respiratory disease) and new statistical methods. Our results showed that discrepancies between human studies and animal studies may reflect errors in the ways human studies were designed and conducted. As a result, these human studies may provide a flawed picture of BCAA effects that could mislead efforts developing novel therapeutics.

[1] Department of Epidemiology, Gillings School of Global Public Health, University of North Carolina at Chapel Hill, Chapel Hill, NC 27516, USA. [2] Carolina Population Center, University of North Carolina at Chapel Hill, Chapel Hill, NC 27516, USA. [3] Department of Biostatistics, Gillings School of Global Public Health, University of North Carolina at Chapel Hill, Chapel Hill, NC 27516, USA. [4] The Charles Bronfman Institute for Personalized Medicine, Department of Environmental Medicine and Public Health, Icahn School of Medicine at Mount Sinai, New York, NY 10029, USA. [5] Department of Genetics, University of North Carolina at Chapel Hill, Chapel Hill, NC 27516, USA. [6] Department of Nutrition, Gillings School of Global Public Health, University of North Carolina at Chapel Hill, Chapel Hill, NC 27516, USA. ✉email: Christy_avery@unc.edu

The three branched chain amino acids (BCAA) leucine, isoleucine, and valine are abundant essential nutrients that account for ~20% of total amino acids in muscle protein[1]. Accurately quantifying individual BCAA effects on metabolism and, more broadly, health and disease, has been challenging because circulating leucine, isoleucine, and valine concentrations are highly correlated. This correlation reflects shared enzymes governing synthesis and degradation as well as dietary patterns in which BCAAs are typically consumed together[2,3]. Despite these high correlations, BCAA intermediates and final metabolites differ, motivating studies examining whether individual BCAAs harbor distinct metabolic effects.

To quantify individual BCAA phenotypic effects, observational studies have examined BCAAs separately. These studies, performed across a wide spectrum of phenotypes, have reported that BCAAs exert homogeneous phenotypic effects[4–13]. However, the observational BCAA literature is inconsistent with experimental animal and human dietary restriction studies, which instead suggest that individual BCAAs exert heterogeneous phenotypic effects[14–16]. Because observational studies cannot isolate the effects of individual BCAAs through dietary manipulation, as is possible in experimental studies, observational studies may be subject to bias from shared genetic and lifestyle factors that govern BCAA intake, synthesis, and degradation. Yet, few BCAA observational studies have examined whether these shared factors bias BCAA phenotypic effects. Observational study designs that can accurately estimate BCAA phenotypic effects are needed because it is infeasible to conduct experimental studies for the potentially large number of phenotypes potentially affected by BCAAs.

Advances in genetics, large-scale biobanks, and causal inference statistical methods offer novel avenues to characterize BCAA effects more accurately. Thus, we leveraged dense phenotypic and genotypic data from the UK Biobank and multivariable Mendelian randomization (MVMR) causal inference methods[17] to simultaneously estimate causal effects of leucine, isoleucine, and valine on 441 phenotypes spanning 17 categories. In contrast to the observational literature demonstrating homogeneity of effect, our results suggested that BCAAs harbored distinct and often opposing causal effects on diverse phenotypes. These findings are consistent with experimental BCAA studies and suggest that the observational literature has provided a highly flawed picture of BCAA phenotypic effects. More broadly, our results motivate the development and application of causal inference methods that enable metabolomics studies, and 'omics studies more generally, to validly estimate phenotypic effects of features with shared lifestyle and genetic factors.

## Methods

**Study participants.** The United Kingdom Biobank (UK Biobank) is a publicly available, longitudinal study of Great Britain residents[18]. Briefly, 9.2 million individuals aged 40–69 years who were registered with the National Health Service and who lived within 25 miles (40 km) of one of 22 assessment centers located in England, Scotland, and Wales were invited. Of these eligible participants, 502,464 participants (5.5%) enrolled in the study. At study baseline (2006–2010), extensive lifestyle, sociodemographic, biologic, and health-related data were collected. This study was approved by the institutional review board (number 19–2281) of the University of North Carolina at Chapel Hill. Informed consent was obtained from all participants. Informed consent was obtained from all participants at study baseline for the lifetime of UK Biobank unless the participant withdraws. Additional consent was not required because consent was sought for research in general rather than for specific research questions.

**BCAA measurement.** Leucine, isoleucine, and valine were measured as part of the simultaneous quantification of 249 metabolite biomarkers. Briefly, a random subset of non-fasting EDTA baseline plasma samples from 118,466 participants were selected. Metabolite biomarker measurement was performed using high-throughput NMR spectroscopy (Nightingale Health Plc; biomarker quantification version 2020)[19,20].

**Phenotypic data.** Prior BCAA observational studies examined associations with cancer[6], cardiovascular[5], metabolic[4,21], cognition[22], and respiratory[23] phenotyps, among others. To enable comparison with this body of literature, we also examined broad phenotypic categories using two data sources: data measured by UK Biobank investigators (investigator-defined phenotypes, n = 249) and phenotypes constructed from inpatient episode records (phecodes, n = 192) (Table 1). For the investigator-defined phenotypes, each of the 257 UK Biobank investigator-defined origin categories was manually reviewed by two authors (CLA and RY or SHK) and flagged if the category included phenotypic data (e.g., blood pressure phenotypes were included, but procedure duration times were excluded). Nutrition/dietary data, blob/bulk data, and data elements measured in fewer than ~5000 participants (as of 01/2021 for data currently being collected) were excluded from further review. Each phenotype was manually reviewed, renamed, cleaned, and classified into a composite phenotype, when appropriate (e.g., angina was classified using Rose's Angina Questionnaire) (Supplementary data 7). Categorical phenotypes with fewer than 200 cases were excluded, as were continuous phenotypes with fewer than 200 observations[24]. For continuous phenotypes we applied an inverse rank normalization when skewness was >|0.75| and excluded outliers outside four standard deviations from the mean. We included prevalent cases rather than incident cases for several diseases (e.g., investigator defined type 2 diabetes and chronic kidney disease) measured at baseline.

For the inpatient episode records, we used International Classification of Diseases, Tenth Revision (ICD-10) codes reported for inpatient episodes from 1997 to 03/2021. ICD-10 codes in the primary position, which represented the condition that was chiefly responsible for the admission, were aggregated into clinically

**Table 1 Overview of 17 phenotype categories examined in an observational study of BCAA phenotypic effects in n = 97,469 European ancestry UK Biobank participants.**

| Category | ICD-10 inpatient phecodes | Investigator-collected phenotypes |
|---|---|---|
| Circulatory system | 24 | 29 |
| Dermatologic | 9 | 0 |
| Digestive | 45 | 0 |
| Endocrine/metabolic | 2 | 43 |
| Genitourinary | 12 | 11 |
| Hematopoietic | 4 | 31 |
| Infectious diseases | 5 | 0 |
| Medications | 0 | 23 |
| Mental and behavioral disorders | 1 | 28 |
| Musculoskeletal | 18 | 16 |
| Neoplasms | 29 | 15 |
| Neurological | 8 | 13 |
| Other | 7 | 0 |
| Reproductive | 3 | 9 |
| Respiratory | 10 | 8 |
| Sense organs | 13 | 23 |
| Symptoms | 2 | 0 |
| Total | 192 | 249 |

relevant diagnosis phecodes based on the ICD-10 coding structure and expert opinion[25] (Supplementary Data 8). Categorical phenotypes with fewer than 200 cases were excluded[24].

**Genotypic data**. UK Biobank participants were genotyped using two very similar arrays: the UK BiLEVE array ($n = 49,950$ participants) or the UK Biobank Axiom array ($n = 438,427$ participants)[26]. The following individuals were excluded: individuals identified by UK Biobank investigators as outliers based on genotype missingness or heterogeneity; individuals whose genotypic sex was discordant with their self-reported sex; and individuals who did not cluster with 1000 Genomes European ancestry populations using k-means clustering with 4 PC dimension due to the small number of non-European ancestry participants with BCAA measurements. Genotypic data were imputed to the Haplotype Reference Consortium reference panel. Missing variants were imputed with the UK 10 K and 1000 G reference panels[26]. Imputed variants with an imputation quality score < 0.4, or effective sample size < 30, calculated as

$$2 \times MAF(1 - MAF) \times N \times Q \qquad (1)$$

where MAF is minor allele frequency, Q is imputation quality, and $N$ is study size, were excluded.

**Multivariable Mendelian Randomization (MVMR)**. We estimated total causal effects of each BCAA on 441 phenotypes (143 continuous and 298 binary) in a maximum of $n = 97,469$ unrelated participants by applying MVMR methods to a causal structure in which the correlation between BCAAs was driven by shared confounders and genetic variants (Fig. 1)[17]. In scenarios such as those specified in Fig. 1, MVMR can be used to estimate the total causal effects of each BCAA. The two-stage least squares (2SLS) and not two-sample MVMR approach was selected because 2SLS enabled smaller variances of the causal parameters of interest, employed test statistics (e.g., the Sanderson-Windmeijer F-statistic) that did not require the knowledge of the covariance between the effect of the instrumental variables on each exposure, and avoided assumptions of source population similarity in two sample calculations[17].

In the first stage, each $X_j = 1,...3$ BCAA was regressed on the BCAA specific PRS[27,28], $G_j$, using linear regression[29] and restricting to unrelated individuals $j = 1,...,N$. Consistent with the 2SLS MVMR framework, BCAA specific PRS included overlapping variants[17]. In the below equation U, $\pi$, and v represented the confounders (e.g., ancestral principal components, study center, age, and sex),

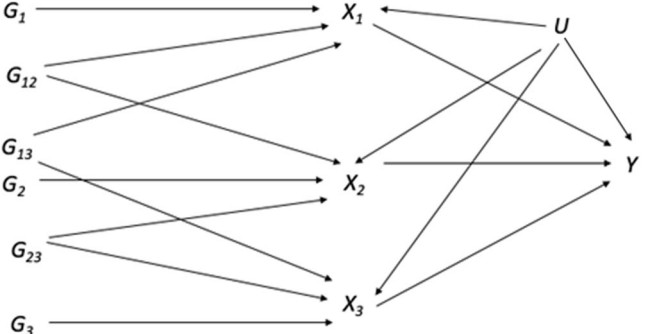

**Fig. 1 Hypothesized relationship between the genetic variants G, branched chain amino acids (BCAAs) $X_1$, $X_2$, and $X_3$, and outcome Y in the presence of unobserved confounder U.** The BCAAs (X) are affected by genetic variants G, with some variants affecting specific BCAAs (e.g., $G_1$) and other genetic variants affecting multiple BCAAs (e.g., $G_{12}$). The unobserved confounder U affects all three BCAAs and outcome Y. Genetic variants G only affect outcome Y through BCAAs X.

parameter vectors, and error terms, respectively. The confounders were identical for all three BCAAs, although confounder parameters were allowed to vary.

$$X_{ij} = \pi_{0j} + \pi_{1j}G_{1j} + \pi_{2j}G_{2j} + \pi_{3j}G_{3j} + \pi_{4i}U_j + v_{ij} \qquad (2)$$

Given acceptable instrument strength (e.g., $F > 10$)[17], the predicted values $\hat{X}_{ij}$ were used in the second-stage regression model to estimate total causal effects of X on Y. In the second stage, each phenotype Y was regressed linearly on the fitted values $\beta_j \hat{X}_{ij}$ estimated in the first stage using either linear regression for continuous traits or probit regression for categorical traits. For simplicity, we provide the two-stage equation predicting the continuous trait, $Y_j$,

$$Y_j = \beta_0 + \beta_{1j}\hat{X}_{1j} + \beta_{2j}\hat{X}_{2j} + \beta_{3j}\hat{X}_{3j} + \beta_4 U_j + v_y \qquad (3)$$

The 2SLS approach was implemented in Stata (StataCorp LP, College Station, Texas) using the *ivreg2* and *ivprobit* commands. Given the goal of this paper was proof of concept rather than hypothesis driven, statistical significance of BCAA total causal effects was determined using a conservative Bonferroni correction accounting for 441 phenotypes and three BCAAs (i.e., $0.05/[441 * 3] = 3.78 \times 10^{-5}$).

Two F tests were used to evaluate instrument strength, a core MVMR assumption: the first F test evaluated whether G strongly predicted individual $X_j$ and the second F test—the Sanderson-Windmeijer F test - evaluated whether G jointly predicted $Xj$ (i.e., once $X_1$ is predicted, G also predicted $X_2$ and $X_3$). While the Sanderson-Windmeijer test is available in *ivreg2*, it is unavailable in *ivprobit*. Therefore, this test was calculated only for the continuous phenotypes. However, except for minor differences in sample size differences due to missing phenotypic data, the first stage models used to generate the Sanderson-Windmeijer *F*-tests were identical regardless of the phenotype distribution (continuous or probit), suggesting that assessments of instrument strength performed for the continuous phenotypes could inform assessments of instrument strength for the categorical phenotypes.

To enable comparison of MVMR total causal effect estimates with estimates that did not account for the correlation between BCAAs, we used univariable MR to estimate the total casual effects of leucine, isoleucine, and valine separately. To provide further context and enable comparison with the published literature, we also estimated phenotypic effects using measured BCAAs as the exposure. Because MVMR is the only approach that accounted for the correlation between BCAAs, we considered MVMR as the least biased approach.

To implement univariate MR, we estimated total casual effects that did not account for the correlation with the other two BCAAs in unrelated individuals using linear regression model in the case of continuous phenotypes and probit regression in the case of binary phenotypes. These models were estimated using SAS (Cary, North Carolina).

**Genome-wide association study (GWAS)**. To enable construction of genetic instrumental variables, we conducted a GWAS on autosomes for each BCAA. We used generalized linear models implemented in SAIGE that accounted for relatedness[30] and took the form

$$M1 : g(E(Y)) = X\alpha + G\beta \qquad (4)$$

where Y was a vector of inverse normalized BCAAs, g was an identity link function, X denoted covariates (age, sex, ancestral PCs, and study center) and G was variant dosage. Variants with P-values < $1.7 \times 10^{-9}$ after genomic control were considered genome-wide significant. Variants that remained significant after stepwise conditional analyses using GCTA (the --cojo-slct

method) with the UK Biobank samples as the LD reference panel were considered independent[31]. Because the PRS were standardized, effects are interpreted per 1 SD increase in PRS.

**Polygenic risk scores (PRS).** Using the GWAS results, PRS were constructed for each BCAA. These PRS serve as the genetic instrumental variables. Because PRS composed of genome-wide significant independent variants explained limited phenotypic variation (partial $R^2$ range: 0.008–1.6%), we first selected a subset of genetic variants by restricting to HapMap3 variants plus all variants GWAS with $P$-values < 0.05, resulting in 1,457,694, 1,499,770, 1,538,890 variants for isoleucine, leucine, and valine include, respectively. Because we did not have independent discovery and target data, we used the Crosspred method to estimate a isoleucine, leucine, and valine PRS for each participant, including the variants described above, as this approach enabled estimation of PRS in single populations using cross-validation to address overfitting[32]. Our PRS are publicly available on the Polygenic Score Catalog (https://www.pgscatalog.org/)[33].

**Narrow-sense heritability.** To estimate the proportion of phenotypic variation in each BCAA captured by the variants (i.e., narrow sense heritability) and approximate an upper bound on the proportion of variation that could be explained by PRS, we used GCTA and restricted to unrelated participants using a King-cutoff in plink of 0.06[34,35]. Briefly, a genetic relationship matrix was created for each chromosome (1–22), including the same variants used to construct the PRS. After combining the 22 matrices, linear models were fit using restricted maximum likelihood in GCTA (GREML method), adjusting for age, sex, ancestral PCs, and study center.

**Reporting summary.** Further information on research design is available in the Nature Portfolio Reporting Summary linked to this article.

## Results

**Study population.** We included $n = 97,469$ participants of European ancestry (Supplementary Data 1). At study baseline, participants were on average middle age (mean age = 56.7 years) and female (54.1%). Approximately 67% of participants were classified as overweight or obese (mean body mass index = 27.4) and 4.6% of participants were classified as having type 2 diabetes. Measured BCAAs were highly correlated ($r$ range: 0.83–0.91), with slightly lower correlations for genetically inferred BCAAs ($r$ range: 0.73–0.80) (Supplementary Data 2).

**GWAS.** Twelve genome-wide significant loci were identified (Fig. 2, Supplementary Data 3). Eight loci were identified for all three BCAAs and valine harbored four additional loci: *GLUD1*, *ACACB*, *MLXIP*, and *PRODH*. Lead variants for these 12 loci showed a wide range in coded allele frequencies (range: 0.002–0.300) and effect per coded allele (absolute value effect size range: 0.033–0.37). BCAA-specific independent signals were identified for several shared loci, including *DDX19A* and *ABCG2*. For the *DDX19A* locus, one independent signal (lead variant rs370014171) was common to all three BCAAs. The second independent signal (lead variant rs2287978) was associated with valine and leucine and the third independent signal (lead variant rs9930957) was associated with leucine. These results demonstrate that the genetic architecture of leucine, isoleucine, and valine has both shared and unique features.

The proportion of phenotypic variance captured by genetic variants ranged from $h^2 = 65\%$ (isoleucine) to $h^2 = 67\%$ (valine) (Supplementary Data 4). BCAA polygenic risk scores (PRS), which

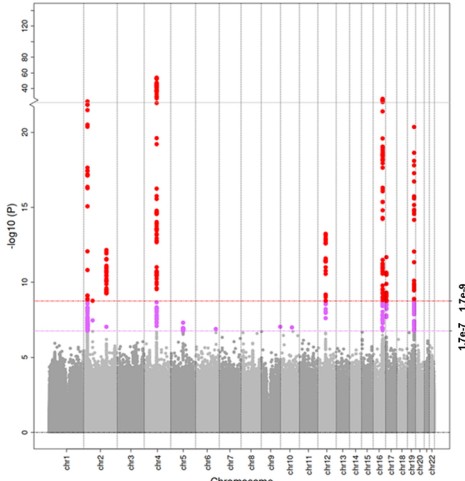

## A. Isoleucine

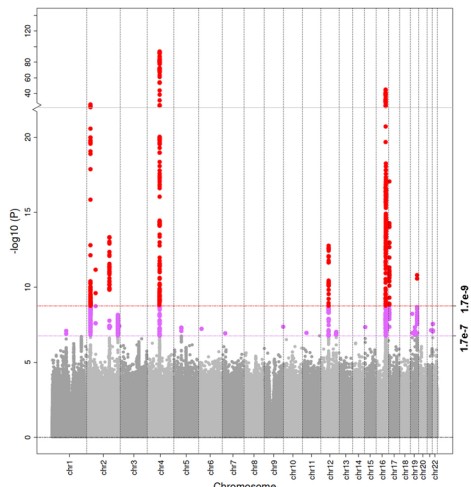

## B. Leucine

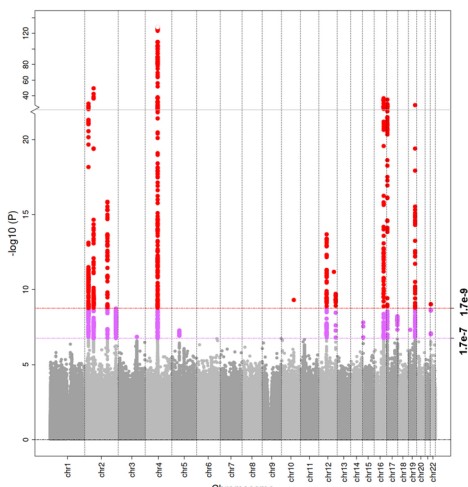

## C. Valine

**Fig. 2 Genome-wide association studies of individual branched chain amino acids in $n = 97,469$ European ancestry UK Biobank participants.** Variant-isoleucine (**A**) variant-leucine (**B**) and variant-valine (**C**) $p$-values are plotted against genome location. Genome-wide significance ($1.7 \times 10^{-9}$) and suggestive ($1.7 \times 10^{-7}$) thresholds are shown in red and pink, respectively.

served as the genetic instrumental variables (IV) for MVMR, explained a slightly smaller proportion (57-58%) of the phenotypic variance in leucine, isoleucine, and valine ($P = 2.2 \times 10^{-16}$).

**MVMR, MR, and observational results**. We then used MVMR, an extended form of univariable Mendelian randomization (MR) developed to estimate causal effects of two or more exposures on an outcome, to estimate BCAA causal effects. Total causal effects are interpreted as the effect of varying one BCAA while keeping the other two BCAAs constant. Because MVMR addresses potential biasing effects of shared genetic and lifestyle factors, this approach provides less biased estimates of BCAA phenotypic effects compared to univariable MR studies or observational studies that do not account for these shared factors[17]. Further, by using genetic IVs, MVMR also is robust to bias from reverse causation[36].

As described in the Methods, MVMR assumes (1) that the BCAA genetic IVs are strongly associated with the exposure[17]. To examine this assumption, individual F-statistics and Sanderson-Windmeijer F-statistics were used to test whether the genetic IVs individually and jointly predicted the exposures. There was sufficient evidence based on the F-statistics (median (range): 36,916 (174, 46762)) and the Sanderson-Windmeijer F-statistics (median (range): 34,847 (156, 45,000)) to suggest that this assumption was not violated (Supplementary Data 9).

Given evidence of strong genetic IVs, we evaluated associations between these IVs with 441 phenotypes across 17 categories (Table 1, Supplementary Data 5). A total of 117 phenotypes across 12 categories (26.5% of available phenotypes) had statistically significant evidence of a total causal effect with at least one BCAA (Fig. 2 outer ring, Supplementary Data 6). Significant total causal effects for phenotypes in the dermatologic, infectious diseases, neoplasms, other, and symptoms categories were not identified.

Of the 117 phenotypes with evidence of a statistically significant total causal effect with at least one BCAA, almost half (44%, $n = 52$) were associated with a single BCAA (Table 2). These 52 phenotypes included total causal effects of valine on diabetic eye disease [odds ratio = 1.51, 95% confidence interval (CI) = 1.31, 1.76], valine on albuminuria (odds ratio = 1.14, 95% CI = 1.08, 1.20), and isoleucine on angina (odds ratio = 1.17, 95% CI = 1.31, 1.76). Only two of these 52 phenotypes (beta blocking agents and drugs affecting bone structure and mineralization) showed evidence of a total causal effect with leucine only. Results from

univariable MR or the analysis of measured BCAAs were largely inconsistent with estimated total causal effects for these 52 phenotypes (Fig. 3, **inner ring**, Table 2, Supplementary Data 6). For example, there were only five phenotypes for which results estimated using univariable MR were concordant in significance and effect direction with total causal effects estimated using MVMR: chronic bronchitis, use of calcium channel blocker medications, current vs. never smoking, monocyte percentage, and neutrophil percentage. Instead, the univariable MR approach estimated statistically significant and directionally consistent BCAA effects for 37 of the 52 phenotypes (71%), including estimated glomerular filtration rate and inguinal hernia. A similarly high proportion of biased estimates and false positive findings for individual BCAA effects was observed when examining phenotypes with significant total causal effects for two BCAAs.

There were 25 phenotypes for which MVMR estimated significant total causal effects for all three BCAAs. For 20 of the 25 phenotypes, statistically significant effects for all three BCAAs also were estimated using univariate MR. However, discordance in effect direction was observed between the two approaches. Total causal effects estimated by MVMR were directionally consistent for only 1 of the 20 phenotypes. In contrast, the univariate MR approach estimated directionally consistent BCAA effects for 19 of the 20 phenotypes.

Finally, discordant findings were observed, whereby the univariate MR approach identified statistically significant findings for 49 phenotypes, but none of the total causal effects estimated by MVMR were significant. These findings may represent scenarios where statistical power to estimate total causal effects via MVMR was reduced. Nonetheless, by accounting for the correlation between BCAAs, the MVMR total causal effects provide a more accurate assessment, even if the P-values were not statistically significant. For example, osteoporosis was uncommon in study participants ($n = 1,142$ cases) and P-values estimated using univariate MR suggested significant and directionally consistent inverse effects for leucine, isoleucine, and valine (P-values < $1.1 \times 10^{-6}$). In contrast, although not statistically significant, MVMR results suggested that isoleucine was positively associated with osteoporosis (P-value = $9.0 \times 10^{-5}$), whereas valine was negatively associated (P-value = $9.7 \times 10^{-5}$).

## Discussion

In this study, we addressed a major limitation in observational studies of BCAAs: the inability to estimate individual effects of leucine, isoleucine, and valine while accounting for shared lifestyle and genetic factors. To address this gap, we used dense phenotypic data from a well characterized biobank and MVMR causal inference methods. We demonstrated distinct and often opposing BCAA phenotypic effects, findings that conflicted with observational studies that did not account for shared lifestyle and genetic factors. These findings suggest that the observational literature examining BCAA phenotypic effects is biased. BCAA observational studies are not only biased regarding direction of effect; any conclusions about the presence (or absence) of a statistically significant association also is prone to substantial error. Such threats to validity are not limited to BCAA studies. All 'omics studies conducted in observational settings should consider the potential for bias when 'omics features are governed by shared factors, although such effects remain largely unrealized.

A large body of observational literature has documented associations between individual BCAAs with human diseases, although few studies have addressed confounding from shared lifestyle and genetic factors. Meta-analyses of cardiovascular disease[37], pre-diabetes and diabetes[21,38], and liver cirrhosis[39] exemplify this practice. For example, a recent meta-analysis reported very precise

**Table 2 Comparison of results from approaches that did (columns) and did not (rows) account for the strong correlation between BCAAs when estimating effects with 441 phenotypes spanning 17 categories in a maximum of 97,469 UK Biobank European ancestry participants.**

| | | No. of phenotypes with significant total causal effect with 0, 1, 2, or 3 BCAAs[b,c] | | | | |
|---|---|---|---|---|---|---|
| | | **0** | **1** | **2** | **3** | **Total** |
| N. of phenotypes with significant total effect with 0, 1, 2, or 3 BCAAs[a,b] | 0 | 275 | 6 | 3 | 1 | 285 |
| | 1 | 13 | 5 | 5 | 1 | 24 |
| | 2 | 9 | 4 | 3 | 3 | 19 |
| | 3 | 27 | 37 | 29 | 20 | 113 |
| | Total | 324 | 52 | 40 | 25 | 441 |

[a]Used univariable Mendelian randomization that did not account for correlation between BCAAs.
[b]All statistical models adjusted for ancestral principal components, study center, age, and sex
[c]Estimated using MVMR statistical approach that accounted for correlation between BCAAs.
BCAA, branched chain amino acid. MVMR, multivariable Mendelian randomization.

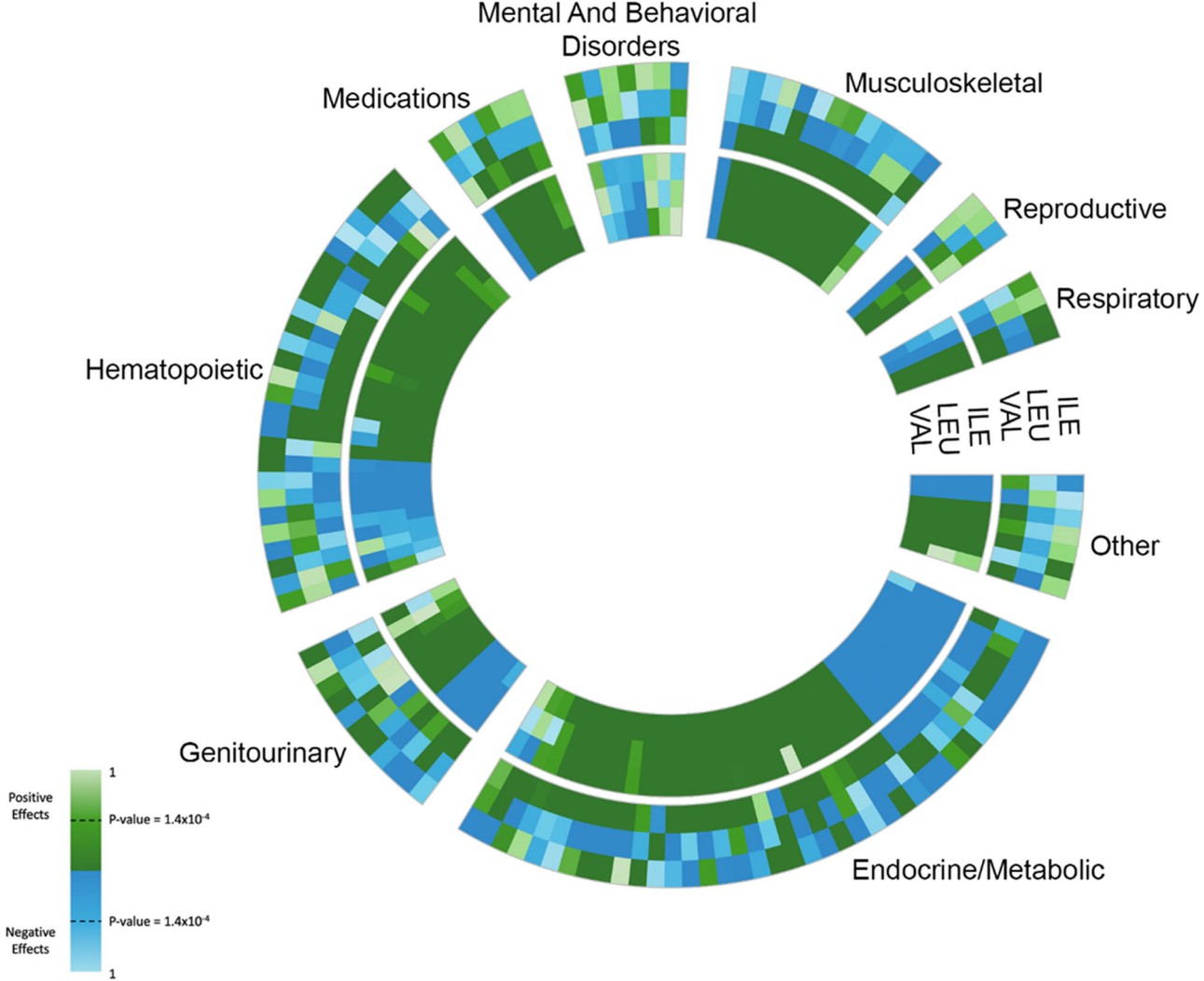

**Fig. 3 Causal inference study of branched chain amino acid (BCAA) phenotypic effects in a maximum of $n = 97{,}469$ UK Biobank European ancestry participants identifies 117 significant associations ($P < 3.78 \times 10^{-5}$) across 17 categories.** In the outer ring, total causal effects and p-values are estimated for each BCAA using multivariate Mendelian randomization, restricting to the 117 phenotypes with evidence of at least one significant total causal effect, and accounting for bias from shared genetic factors. In the inner ring, total causal effects and p-values are estimated for each BCAA using univariable Mendelian randomization methods that do not account for bias from shared genetic factors. In each ring, isoleucine (ILE) is in the outermost position, leucine (LEU) is in the middle position, and valine (VAL) is in the internal position. All statistical models adjusted for ancestral principal components, study center, age, and sex.

and highly significant relative risks for one standard deviation increases in isoleucine, leucine, and valine with type 2 diabetes of 1.54 (95% confidence interval [CI]: 1.36, 1.74), 1.40 (95% CI: 1.29, 1.52), and 1.40 (95% CI: 1.25, 1.57), respectively[21]. However, because none of the 19-to-23 included studies accounted for shared genetic factors, these estimates are biased. This bias constrains inference on the presence, direction, and magnitude of effect. Similarly, univariable MR causal inference studies[40] could not identify which of the BCAAs had causal effects on type 2 diabetes[41].

In contrast, by estimating BCAA total causal effects, our results suggested a different picture: that valine significantly increased type 2 diabetes, leucine significantly decreased type 2 diabetes, and isoleucine had no significant effect. It is biologically plausible that each BCAA harbors distinct effects on type 2 diabetes. For example, each BCAA differentially activates the mechanistic target of rapamycin complex 1 (mTORC1), a central regulator of cell growth and metabolism[42]. Intermediate and final end-products of BCAA metabolism also differ, including 3-hydroxyisobutyrate, a valine metabolite that promotes muscle lipid accumulation and insulin

resistance in mice[16]. Similarly, leucine is exclusively ketogenic and prior research has demonstrated that ketone bodies can improve insulin sensitivity and attenuate insulin resistance[43,44]. Discrepancies observed for type 2 diabetes parallels our overall finding, highlighting the potential for biased inference regarding effect presence vs. absence and effect direction when the correlation between BCAAs was ignored. At best the assumed homogeneity of effects may lead to ineffective therapies. At worst, assumed homogeneity may lead to unintended impacts where combined BCAA therapies have directionally opposing effects.

Experimental studies also have demonstrated that individual BCAAs have different catabolites and catabolic intermediates that likely exert distinct effects on metabolism, providing further support for our findings. In one example, isoleucine or valine restriction, but not leucine restriction, restored metabolic health in diet-induced obese mice, even when the animals consumed Western diets[14]. Another study examining metabolic consequences of high fat diets in mice was more consistent with our findings for type 2 diabetes. Specifically, this study demonstrated

that leucine supplementation had beneficial effects on adiposity and insulin sensitivity, whereas valine supplementation reduced glucose tolerance and insulin sensitivity[15]. It is challenging to compare these findings with our results, given the UK Biobank's observational design. In addition to design differences, the metabolic processes influencing weight (e.g., body mass index) and long-term glycemic control (e.g., glycated hemoglobin), which are available in the UK Biobank, also may differ from the metabolic processes influencing weight loss and glucose metabolism regulation, which were not measured[45].

Despite several strengths, our study is not without limitations. One challenge is disentangling intermediate phenotypes directly affected by individual BCAAs from downstream phenotypes affected by the intermediate phenotypes (i.e., vertical pleiotropy). For example, several phenotypic effects may represent downstream effects of insulin resistance. Although outside the scope of this effort, mediation analysis methods that build upon the strengths of MVMR are emerging and may inform future studies[27,46]. Second, we used one sample MVMR, thereby limiting our analysis to a subset of UK Biobank participants with genotypic, phenotypic, and BCAA data. Although two sample MVMR methods may have enabled a larger and more powerful study, benefits of one sample MVMR included smaller variances of causal parameters, application of test statistics (e.g., the Sanderson-Windmeijer F-statistic) that did not require knowledge of the covariance between the effect of the instrumental variables on each exposures, and avoidance of assumptions of source population similarity when using non-overlapping populations[17]. Given the large sample sizes available in the UK Biobank, we felt that one sample strengths outweighed the limitations. Third, we focused on curated phenotypes, which yielded 441 phenotypes of relatively high accuracy across 17 categories. Although our approach may have omitted some relevant phenotypes, especially given our choice of a conservative multiple comparisons adjustment, a comprehensive cataloging of BCAA phenotypic effects was not our main goal. Instead, our goal was to examine whether discrepancies between experimental and observational BCAA studies reflected bias from shared genetic and lifestyle factors. Fourth, challenges were potentially introduced by the potential for horizontal pleiotropy, whereby the genetic IV affected the outcome independent of its effect on the BCAAs. This bias cannot be tested directly, but instead is gauged using sensitivity analyses that are based on additional assumptions. Few methods robust to horizontal pleiotropy have been developed for 2SLS MVMR and even fewer exist when PRS genetic IVs are used[47]. Although horizontal pleiotropy may introduce some degree of bias in the total causal effects, the fact that we are simultaneously modeling the major source of horizontal pleiotropy (i.e., the three BCAAs), the magnitude of effects, the very low levels of significance, and the consistency of our main finding – that BCAAs exert unique effects across diverse phenotypes that would be subject to different levels of horizontal pleiotropy- together suggest that the potential for horizontal pleiotropy changing our main conclusions is low.

In conclusion, contrary to the published observational literature, our findings suggest considerable phenotypic heterogeneity in BCAA effects. These findings emphasize the need for future observational studies that systematically examine the causal architecture of correlated phenotypes, including 'omics and other sources of big data. These investigations may be challenging, as while the main pathways of BCAA catabolism are relatively well-characterized, for other questions the exposures may be unknown, or the main pathways may be poorly characterized. Surmounting these challenges will undoubtedly require causal inference innovations applicable to observational settings. These innovations will help interpret conflicting literature, improve the quality of available evidence, illuminate disease pathogenesis, and, ultimately, aid the development of effective treatments, biomarkers, and prediction algorithms for public health and clinical action.

## Data availability

The UK Biobank phenotypic and genotypic data are publicly available https://www.ukbiobank.ac.uk/enable-your-research/register. Sharing of the UK Biobank data is governed by the parent study. Supplementary Data 3 and Supplementary Data 6 contain source data for Fig. 2 and Fig. 3, respectively. Additional data for the supplementary figures are available upon request.

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

## Acknowledgements

This work was supported by UK Biobank application 25953. The following grants supported this study: R01HL147853 (Avery), R01HL142828 (Avery, Ballou, Howard, North), R01HL151152 (Avery, Graff, North), R01HG010297 (Avery, Graff, North), R01HL143885 (Avery, Gordon-Larsen, Howard, North), T32HL007055 (Lee), T32HL129982 (Downie), KL2TR002490 (Raffield), and F32HL149256 (Lee).

## Author contributions

C.L.A., A.G.H., K.E.N., P.G.L., and M.G. conceived of and designed the study. C.L.A., A.G.H., H.H.L., C.G.D., M.P.L., S.H.K., A.F.B., M.P., L.M.R., R.Y., and M.G. acquired, prepared and/or analyzed the data. C.L.A., A.G.H., K.E.N., P.G.Y., and M.G. wrote the manuscript. All authors discussed the results and commented on the manuscript.

## Competing interests

The authors declare no competing interests.
