## [Peer Review File · Communications Medicine]

Reviewers' comments:

Reviewer #1 (Remarks to the Author):

This is a very interesting study by Avery and colleagues. I am not qualified to review the statistical approach and models here, but conceptually the idea that different BCAAs have distinct effects on disease and metabolic phenotypes is consistent with the growing number of articles showing that feeding rodents diets with altered levels of one of the BCAAs has amino acid-specific effects. So this is a very timely article and quite intriguing.

I think the main question I have is why the results shown here do not align with the findings of Yu et al, 2021 cited by the authors, which show that isoleucine is positively associated with adiposity and fasting blood glucose and glucose tolerance in mice and with BMI in humans; as well as Deelen et al 2019, which found isoleucine associated with mortality. It occurs to me that maybe sex was not considered in the model; if there are effects in only one sex or the effects of isoleucine or valine or leucine were significant in only one sex, or different in each sex, would the model detect that?

Finally, Table S1A-H is nice, but the authors should use colors or fonts to highlight the adjusted significance hold

Reviewer #2 (Remarks to the Author):

This is an interesting phenome wide association study (Phewas) looking at the direct effect of three linked BCAAs (leucine, valine and isoleucine) on phenotypes from the UK biobank using multivariable mendelian randomization, so it could make an interesting contribution. However, there seem to be some key points that worry me about validity of the design and analysis

- 1) The 3 BCAAs are highly correlated, so it is important to know if there is enough variability between for them to be considered together and to obtain valid estimates. The conditional F-statistics give an estimate of the strength of each instrument (for each exposure) considering the others. I can't find the conditional F-statistics for each exposure in the paper. So, I estimated them myself and they were all below the "rule of thumb" cut-off of 10, indicating possible weak instrument bias and invalid estimates.
- 2) The Phewas is based on 257 investigator defined phenotypes and 184 phenotypes based on inpatient hospital records from the UK biobank. Most Phewas of the UK Biobank use several thousand phenotypes. How comprehensive are these phenotypes?
- 3) The paper gives the direct effects of three linked BCAAs but does not give the underlying causal structure between the BCAAs, so it is not clear that these are all direct effects. For example, consider an analysis of leucine and valine on T2DM. If leucine causes valine and valine causes T2DM then an analysis of leucine on T2DM allowing for valine gives the direct effect of leucine on T2DM. However, if valine causes leucine and T2DM then an analysis of leucine on T2DM gives the total effect of leucine on T2DM. The authors also write about correlation between the BCAAs, when it would help to start by explaining the causal relationships between all 3 BCAAs.

Minor comments

Summary

It is not quite clear to me why causal inference helps with clarifying correlations.
Please say how many phenotypes were considered.

Introduction

Please explain in a bit more detail how leucine, isoleucine and valine are causally related.

Methods

Study description

The UK currently includes Northern Ireland, the UK Biobank did not recruit residents of Northern Ireland. So, the UK Biobank recruited from Great Britain, please amend accordingly.

The 9.2 million aged 40-69 registered with the NHS are too few to be all the 40-69 years olds in Great Britain. Is it just people who lived within a certain distance of the 22 assessment centres who were in the UK Biobank sampling frame?

What is the rationale for excluding values more than four standard deviations from the mean? In such a large sample these could be valid measurements.

Did the calculation of the multivariable F-statistics take into account the covariance between the exposures? If not, please explain how this affects the F-statistic.

Please justify the choice of using prevalent cases rather than incident cases for some phenotypes.

Please explain the consideration of multiple comparisons

Results

Please give the causal structure between leucine, valine and isoleucine, so as to contextualize and help interpret the findings.

Discussion

It would be helpful to explain in more detail how well the assumptions of MVMR were met.

Referee expertise:

Referee #1: BCAAs & health/disease

Referee #2: MR, epidemiology

Reviewers' comments:

Reviewer #1 (Remarks to the Author):

This is a very interesting study by Avery and colleagues. I am not qualified to review the statistical approach and models here, but conceptually the idea that different BCAAs have distinct effects on disease and metabolic phenotypes is consistent with the growing number of articles showing that feeding rodents diets with altered levels of one of the BCAAs has amino acid-specific effects. So this is a very timely article and quite intriguing.

I think the main question I have is why the results shown here do not align with the findings of Yu et al, 2021 cited by the authors, which show that isoleucine is positively associated with adiposity and fasting blood glucose and glucose tolerance in mice and with BMI in humans; as well as Deelen et al 2019, which found isoleucine associated with mortality. It occurs to me that maybe sex was not considered in the model; if there are effects in only one sex or the effects of isoleucine or valine or leucine were significant in only one sex, or different in each sex, would the model detect that?

Thank you to the reviewer for their encouraging review. We address each question below.

Comparisons with experimental animal findings: We also were intrigued by the comparison of our findings with findings from Yu et al. Our interpretation of the major findings reported by Yu et al. are that *dietary restriction* of both isoleucine and valine improve glucose tolerance and blunted weight gain, whereas *dietary restriction* of leucine showed no effect. Other recent literature, including Bishop et al., 2022, demonstrated that under *high fat diets*, beneficial effects of leucine and detrimental effects of valine on obesity and insulin were observed. Isoleucine was not examined. We believe these studies are largely consistent with our findings, as both report detrimental effects of valine. Potential drivers of the inconsistencies include the nutritional status of the animals (or humans), dietary composition or other nutrients, and the interventions. Further, the genetic instrumental variables capture ~long-term BCAA levels, whereas the experimental interventions introduce short-term changes. As we describe on line 252, an important next step would be to examine the effects of BCAAs in existing trials with dense genotypic data and BCAA measurements.

Comparisons with observational human findings: The studies performed by Yu et al. and Deelen et al, while interesting, highlight the challenges of studying BCAAs in observational settings. Briefly, Yu et al., perform a cross-sectional study where BCAA intake was inferred from questionnaire data and each BCAA was evaluated in a separate statistical model. It is difficult to interpret these findings, as they are likely subject to reverse causation from the cross-sectional design and bias, as described in our updated introduction (e.g., line 33). We selected

multivariable Mendelian randomization because this approach is robust to reverse causation and bias-related threats to validity.

Although our work focused on estimating effects whereas Deelen et al., examined prediction, both studies faced challenges introduced by highly correlated BCAAs. Previous simulation studies have demonstrated that very high correlation between predictors can affect both the magnitude and direction of regression coefficients.¹ It is difficult to compare our results with those reported by Deelen et al., because Deelen did not utilize statistical models that could address the high correlation between BCAAs, potentially introducing bias in estimated coefficients. This bias can change both the magnitude and direction of estimated regression coefficients.

Please see line 42 where we include the Bishop et al reference as well as a more detailed discussion of the challenges comparing our results to the experimental results of the observational studies described above. We also modified the introduction to add more clarity in the connection between correlation, bias, and causal inference.

Sex-specific effects: The author raises an interesting question about the role of sex. All statistical models adjusted for sex and thus calculated “sex-averaged” effects. We chose not to test for sex-specific effects, except for phenotypes like free testosterone, sex hormone binding globulin, and ovarian cancer for which *a priori* stratification by sex or restriction to a specific sex was warranted. This decision reflected the already large scope of our study. Nonetheless, for free testosterone and sex hormone binding globulin, we saw patterns that paralleled our overall finding, i.e., heterogeneous phenotypic effects when considering the three BCAAs simultaneously, but biased homogeneous phenotypic effects when considering each BCAA individually (see **Appendix Table 6**). Ovarian cancer was not associated with BCAAs. Exploring potential differential effects of BCAAs by sex or other environmental exposures could be important avenues for future work.

Finally, Table S1A-H is nice, but the authors should use colors or fonts to highlight the adjusted significance hold

We now highlight the significant finds in Appendix Table 3 and Appendix Table 6.

Reviewer #2 (Remarks to the Author):

This is an interesting phenome wide association study (Phewas) looking at the direct effect of three linked BCAAs (leucine, valine and isoleucine) on phenotypes from the UK biobank using multivariable mendelian randomization, so it could make an interesting contribution. However, there seem to be some key points that worry me about validity of the design and analysis

1) The 3 BCAAs are highly correlated, so it is important to know if there is enough variability between for them to be considered together and to obtain valid estimates. The conditional F-statistics give an estimate of the strength of each instrument (for each exposure) considering the others. I can't find the conditional F-statistics for each exposure

in the paper. So, I estimated them myself and they were all below the “rule of thumb” cut-off of 10, indicating possible weak instrument bias and invalid estimates.

Our revised manuscript describes our efforts to ensure the validity of our study design and analysis. Please see line 131 where we provide the range of the conditional F statistics for each BCAA. As shown in the manuscript and in our new Appendix Table 9, these F statistics all exceed the “rule of thumb” threshold of 10. Please note that the statistical program used - STATA - does not provide conditional F statistics for probit models of binary phenotypes. However, the F statistics estimated for the continuous phenotypes ranged from 156 to 45,000. All of these F statistics exceeded the recommended threshold of 10, many by several orders of magnitude. This observation, as well as the fact that the conditional F statistics rely on results from the first stage, a stage common across all phenotypes regardless of outcome distribution, suggested that it was reasonable to assume there was little evidence of weak instrument bias for binary phenotypes.

Although it is difficult to pinpoint the exact reason for the discrepancies between our analyses and the reviewer’s analysis, one possible reason might be the genetic instruments used. In this paper we used BCAA polygenic risk scores (PRS) as our genetic instruments. These PRS summarized all variant effects for each BCAA into a single score, yielding a PRS for each of three BCAAs. As demonstrated by Pierce et al. and Burgess et. al, PRS can provide valid causal estimates and can help address violations of the weak instrument variable assumptions, albeit with potential losses in statistical power.^{2,3} We now include these two references when describing MVMR. We also now describe (line 440) how we will make our BCAA PRS publicly available to enable additional evaluation and causal inference studies.

To further investigate the reviewer’s concern, we also examined genetic instrument performance when instruments were constructed using individual variants rather than a PRS. This approach seemed to mirror the reviewer’s approach, i.e., this approach yielded similarly small F-statistics. Again, this finding suggests supports the validity of our study design and analysis.

2) The Phewas is based on 257 investigator defined phenotypes and 184 phenotypes based on inpatient hospital records from the UK biobank. Most Phewas of the UK Biobank use several thousand phenotypes. How comprehensive are these phenotypes?

We evaluated fewer phenotypes than previous Phewas because we prioritized high quality curated phenotypes over a comprehensive approach using phenotypes with limited curation. As we now discuss (beginning on line 263) “Taking a rigorous approach, we intentionally focused on curated phenotypes that were manually constructed or categorized using phecodes, which yielded 441 phenotypes of relatively high accuracy across 17 categories. This approach contrasts with other “phenome-wide” approaches that include thousands, not hundreds, of phenotypes with little curation.³⁴ Our approach is unique, as few studies have contrasted inferences obtained from studies of curated phenotypes, which offer higher accuracy and statistical power relative to the more comprehensive but less accurate approach using uncurated phenotypes. A curated approach also allows more sensitivity, as phenotypes that were unlikely to be affected by BCAAs were excluded (e.g., variables describing the type of bread eaten yesterday). Although we appreciate that our approach may have omitted some important phenotypes, our main finding

of heterogeneous BCAA effects was observed for phenotypes within 12 of the 17 categories, suggesting broad assessment of BCAA phenotypic effects.”

3) The paper gives the direct effects of three linked BCAAs but does not give the underlying causal structure between the BCAAs, so it is not clear that these are all direct effects. For example, consider an analysis of leucine and valine on T2DM. If leucine causes valine and valine causes T2DM then an analysis of leucine on T2DM allowing for valine gives the direct effect of leucine on T2DM. However, if valine causes leucine and T2DM then an analysis of leucine on T2DM gives the total effect of leucine on T2DM. The authors also write about correlation between the BCAAs, when it would help to start by explaining the causal relationships between all 3 BCAAs.

This suggestion is excellent. We now include a directed acyclic graph illustrating the hypothesized causal relationship (Figure 3). The G in the causal diagram reflect variants governing shared metabolic pathway, which also are now discussed in the introduction (line 37).

Minor comments

Summary

It is not quite clear to me why causal inference helps with clarifying correlations. Please say how many phenotypes were considered.

Our summary has been revised to describe the connection between causal inference and correlations: “The three branched chain amino acids (BCAA) leucine, isoleucine, and valine are essential nutrients that have been associated with numerous diseases, including diabetes, cancer, and cardiovascular disease. Most observational studies suggest that BCAAs exert homogeneous phenotypic effects. These studies may be biased because the three BCAAs are highly correlated, and few studies have accounted for this correlation. Using data from a large and well-characterized biobank and causal inference methods developed for correlated exposures, we estimate distinct and often opposing BCAA causal effects for 117 of 441 phenotypes spanning circulatory, endocrine, and genitourinary categories, among others. Our results suggest that the observational literature has provided a highly flawed picture of BCAA exposure effects that could mislead efforts developing novel therapeutics and interventions targeting BCAAs. More broadly, these findings motivate the development and application of causal inference approaches that enable ‘omics studies to account for the distorting effects of highly correlated exposures..”

As shown above, we now include how many phenotypes were considered and describe the connection between causal inference and BCAA correlation.

Please note that we also updated the introduction to clarify connections between correlation and causal inference (line 35).

Introduction

Please explain in a bit more detail how leucine, isoleucine and valine are causally related.

We now include a sentence in the introduction describing how BCAAs have shared enzymes governing synthesis and degradation (line 37). We also include a directed acyclic graph, as described above (Figure 3).

Methods

Study description

The UK currently includes Northern Ireland, the UK Biobank did not recruit residents of Northern Ireland. So, the UK Biobank recruited from Great Britain, please amend accordingly.

The amendment has been made (line 308).

The 9.2 million aged 40-69 registered with the NHS are too few to be all the 40-69 years olds in Great Britain. Is it just people who lived within a certain distance of the 22 assessment centres who were in the UK Biobank sampling frame?

This amendment also has been made (line 309).

What is the rationale for excluding values more than four standard deviations from the mean? In such a large sample these could be valid measurements.

We erred on the side of caution when using the 4SD exclusion criteria, but upon further consideration agree with the reviewer that this approach could exclude real measurements. In future studies we will consider 4.5 SD. In the context of the present work, on average 0.25% of continuous phenotype values fell outside $> |4|$ SD and categorical results were unaffected. Thus, we anticipate that this oversight had little influence on study findings.

Did the calculation of the multivariable F-statistics take into account the covariance between the exposures? If not, please explain how this affects the F-statistic.

Please see our response above where we provided additional details on calculation of the F-statistics. To reiterate and provide additional clarity, we calculated two separate F-statistics for each BCAA, one F-statistics from the first stage models which did not account for covariance between exposures, and the Sanderson-Windmeijer F-statistic, which accounted for the covariance between exposures. We now include both F statistics in Appendix Table 9 as they both provide important information with regards to the MVMR assumptions.

Please justify the choice of using prevalent cases rather than incident cases for some phenotypes.

We now justify our choice of using prevalent rather than incident cases as follows (line 342): “We included prevalent cases rather than incident cases for several diseases (e.g., investigator defined type 2 diabetes and chronic kidney disease) because longitudinal data often were unavailable on the entire cohort to measure incidence.” Please note that we measured incidence for several endpoints, including cancer endpoints (see Appendix Table 7). We prioritized estimating incidence for cancer endpoints, among others, given anticipated poor capture of

cancer in the hospital inpatient data.

Please explain the consideration of multiple comparisons

Statistical significance of BCAA direct causal effects estimated using the 2SLS approach was determined using Bonferroni correction accounting for 441 phenotypes and three BCAAs (i.e., $0.05/[441*3] = 3.78 \times 10^{-5}$) (line 389).

Results

Please give the causal structure between leucine, valine and isoleucine, so as to contextualize and help interpret the findings.

Please see our new Figure 3, as well as modified introduction.

Discussion

It would be helpful to explain in more detail how well the assumptions of MVMR were met.

Please see our updated discussion, particularly where we discuss horizontal pleiotropy (line 283), likely the largest challenge faced by MR studies.

REFERENCES

1. Vatcheva KP, Lee M, McCormick JB and Rahbar MH. Multicollinearity in Regression Analyses Conducted in Epidemiologic Studies. *Epidemiology (Sunnyvale)*. 2016;6.
2. Burgess S and Thompson SG. Use of allele scores as instrumental variables for Mendelian randomization. *Int J Epidemiol*. 2013;42:1134-44.
3. Pierce BL, Ahsan H and Vanderweele TJ. Power and instrument strength requirements for Mendelian randomization studies using multiple genetic variants. *Int J Epidemiol*. 2011;40:740-52.

Reviewers' comments:

Reviewer #1 (Remarks to the Author):

I think the conclusions here are too strong, and subject to their own forms of bias. In fact, until recently most studies have inappropriately grouped the three BCAAs together, and considering these amino acids separately is required to understand them.

1) One issue - if dietary BCAAs group together, the same is probably generally true for all dietary essential amino acids, as abundance of all nine is probably driven by protein content. Is the correlation of BCAA levels in foods higher than that of any three random essential amino acids?

2) Fundamentally - and not discussed by the authors - valine and leucine results in different catabolites and distinct catabolic intermediates that likely have distinct effects on metabolism.

3) It is clear from animal studies that the individual BCAAs have distinct effects; for example Jang et al 2016, Nature Medicine, showed that the valine specific catabolites 3-HIB has specific effects on fatty acid uptake. While Yu et al 2021 Cell Metab showed that isoleucine and valine restriction have opposite effect from leucine restriction on body composition and glucose tolerance. These are well-controlled animal studies not subject to the types of biases discussed by the author.

4) Further, the Deleen et al 2019 study - an unbiased machine learning study - showed that blood isoleucine was positively associated with mortality, while blood leucine and valine were negatively associated with mortality. It seems to me that if the correlation of these three BCAAs was causing bias, the bias should be in the same direction for all three BCAAs.

Overall I feel this is an important article but that the conclusions here do not adequately address the growing biological evidence that these amino acids have distinct metabolic and physiological effects and consequences.

Reviewer #2 (Remarks to the Author):

1. The conditional F-statistics, that have been added, are very helpful. They look quite high, so it would be very helpful to clarify a couple of points about how the multivariable instruments and conditional F-statistics were generated. First, as regards the multivariable instruments, my understanding is that multivariable instruments should exclude genetic variants correlated across exposures. In addition, the cutoff for genetic instruments is typically a genome wide association with the exposure. Please could the authors provide clarification on these points, justification for the relevant analytic choices and crucially multivariable results excluding genetic variants correlated across exposures and only including genome wide significant genetic variants. Second, the conditional F-statistic should include the covariance matrix between the exposures, when the exposures come from the same data source (<https://pubmed.ncbi.nlm.nih.gov/30535378/>), as here. Please clarify whether any such covariance matrix was used and if not please explain the implications for the multivariable F-statistics. Third, please ensure that the number of variants for each exposure is included in the multivariable analysis given in the results section of the main text.

2. Phewas studies are intended to be comprehensive, so the restriction to a relatively small number

of curated phenotypes does not seem consistent with conducting a Phewas. It would make more sense to explicitly restrict the Phewas to items that could be the result of BCAAs. It would also be very helpful to give a definition of 'curated' and then explain how it corresponds to other criteria for conducting a valid mendelian randomization (MR) study, such as avoiding covariable adjustment, and for conducting valid studies of causal inference, such as counting exposure time correctly. For example, reports of genetically predicted reported medication use generate a lifelong exposure when medication use is not usually life long. It would be very helpful to give results for a comprehensive set of phenotypes, selected on criteria concerning the relevance and validity of each available item. It would also be helpful to validate the findings using other data sources for the outcomes.

3. The paper aims to present the direct effects of three BCAAs. A direct effect is the effect of exposure on outcome absent the mediator, i.e., after accounting for a mediator. This means that the authors are postulating that the effect of each BCAA could be mediated by the other two BCAAs, which is not possible, as explained previously. Moreover, the new Figure 3 showing the analytic structure does not show any mediation, instead it looks like the authors are thinking that the three BCAAs do not cause each other but may be subject to genetic pleiotropy. Specifically, the new Figure corresponds to Figure 4 scenario 3 of this paper laying out the questions addressed by multivariable MR (<https://pubmed.ncbi.nlm.nih.gov/30535378/>). The paper needs to be rewritten either as assessing the independent effect of each BCAA, assuming genetic pleiotropy only along with physiological justification for this approach, or some other justifiable and coherent underlying structure linking the three BCAAs needs to be specified and tested. Logically, it cannot be the current presentation where it appears that each BCAA mediates the other two. All relevant language throughout the text needs to be amended to reflect the question addressed

4. This is a one-sample MR study, which at the limit equates to a two-sample study. However, biases are possible in smaller studies, which should be discussed here.

5. The explanation for using prevalent cases, i.e., that longitudinal data is often unavailable for the entire cohort, is confusing. The UK Biobank participants were followed up through electronic record linkage to death records, cancer registries, primary care and hospital in-patient records <https://www.ukbiobank.ac.uk/explore-your-participation/following-your-health>. The UK has a national health service (NHS) with very limited provision of care outside the NHS. So, record linkage to care within the NHS and to death records should be fairly complete. Please can this point be clarified.

Reviewers' comments:

Reviewer #1 (Remarks to the Author):

I think the conclusions here are too strong, and subject to their own forms of bias. In fact, until recently most studies have inappropriately grouped the three BCAAs together, and considering these amino acids separately is required to understand them.

Thank you very much for re-reviewing our manuscript. We appreciate that time is precious, and you have many competing priorities. Again, thank you for helping increase the quality of our submission.

We completely agree with the reviewer that it is important to understand the separate effects of individual BCAAs. Rereading our manuscript, we believe that confusion was introduced by our failure to precisely distinguish between the observational and experimental literature. Please see changes throughout the manuscript (abstract through discussion), where we describe how estimating separate BCAA effects *in observational studies* requires tailored statistical models that account for the correlation in BCAAs induced by a shared genetic architecture governing metabolism as well as strong confounding by factors like diet. Please note that although the statistical models include all three BCAAs simultaneously to account for the strong correlation, this approach was developed to estimate separate effects of each BCAA.

1) One issue - if dietary BCAAs group together, the same is probably generally true for all dietary essential amino acids, as abundance of all nine is probably driven by protein content. Is the correlation of BCAA levels in foods higher than that of any three random essential amino acids?

We focused on BCAAs because of the large literature examining BCAAs together and the shared pathways governing BCAA metabolism. It would be difficult to use estimates of correlation in protein content to infer correlations in circulating amino acids. For example, dietary intake and requirements vary by amino acid (<https://www.ncbi.nlm.nih.gov/books/NBK234922/table/ttt00012/?report=objectonly>). The digestibility of protein sources also varies by food and presence of other nutrients like fiber. These and other factors likely underlie – at least partially – variation in circulation concentrations of amino acids in human populations (<https://www.nature.com/articles/ejcn2015144>). However, this question does echo a larger theme of our manuscript, e.g., the challenges induced by correlated data in observational settings.

2) Fundamentally - and not discussed by the authors - valine and leucine results in different catabolites and distinct catabolic intermediates that likely have distinct effects on metabolism.

Thank you for identifying this omission. Please see lines 39-42 and lines 193-197 where we discuss the distinct effects catabolic intermediates likely have on metabolism. As described above, we are not arguing for the absence of distinct metabolic effects. We are arguing that current (biased) analytic approaches used to evaluate BCAA effects *in observational studies* erroneously suggest homogenous effects. As described in our response above, we have modified our manuscript significantly to add clarity to this point.

3) It is clear from animal studies that the individual BCAAs have distinct effects; for example Jang et al

2016, Nature Medicine, showed that the valine specific catabolites 3-HIB has specific effects on fatty acid uptake. While Yu et al 2021 Cell Metab showed that isoleucine and valine restriction have opposite effect from leucine restriction on body composition and glucose tolerance. These are well-controlled animal studies not subject to the types of biases discussed by the author.

Please see our response above.

4) Further, the Deleen et al 2019 study - an unbiased machine learning study - showed that blood isoleucine was positively associated with mortality, while blood leucine and valine were negatively associated with mortality. It seems to me that if the correlation of these three BCAAs was causing bias, the bias should be in the same direction for all three BCAAs.

Because our study design differed from the design used by Deelen et al. and we did not examine all-cause mortality, we limited our description of this study to a single brief sentence (line 46) cataloging the existing literature examining phenotypic effects in human populations. We otherwise did not comment on the magnitude or direction of bias in the manuscript (although we did offer some comments in our prior response). We apologize if we suggested that our comment reflected actual changes in our manuscript.

Overall I feel this is an important article but that the conclusions here do not adequately address the growing biological evidence that these amino acids have distinct metabolic and physiological effects and consequences.

We hope our revisions above have addressed the reviewer's concerns about the growing biologic evidence suggesting distinct metabolic and physiological effects. However, identification of these distinct effects in observational studies requires a statistical framework that is largely absent in the published literature.

Reviewer #2 (Remarks to the Author):

1. The conditional F-statistics, that have been added, are very helpful. They look quite high, so it would be very helpful to clarify a couple of points about how the multivariable instruments and conditional F-statistics were generated. First, as regards the multivariable instruments, my understanding is that multivariable instruments should exclude genetic variants correlated across exposures. In addition, the cutoff for genetic instruments is typically a genome wide association with the exposure. Please could the authors provide clarification on these points, justification for the relevant analytic choices and crucially multivariable results excluding genetic variants correlated across exposures and only including genome wide significant genetic variants.

Thank you very much for re-reviewing our manuscript. We appreciate that time is precious, and you have many competing priorities. Again, thank you for helping increase the quality of our submission.

Exclusion of correlated genetic variants. This assumption, while required for univariate MR, is assumed to be violated for MVMR, as stated in Sanderson et al., 2019. Specifically, when describing correlated variants, the authors state that MVMR “estimate(s) a different causal effect and provide(s) benefits

when in fact some of the SNPs may have effects on more than one exposure (thus making them invalid instruments for a univariable MR analysis).” To add additional clarity, we now state that a strength of MVMR is the ability to accommodate variants that have effects on more than one exposure (lines 330-331). Because of this feature of MVMR, we do not present results excluding genetic variants correlated across exposures.

P-value criterion for SNPs included in PRS. Evolving research has demonstrated that for many complex phenotypes, particularly complex phenotypes with a polygenic architecture, PRS accuracy is maximized when a more liberal SNP P-value threshold is applied (<https://pubmed.ncbi.nlm.nih.gov/20562875/>). We now state that our p-value threshold was in part motivated by the poor performance of genetic instruments composed of genome-wide significant loci, e.g., “Because PRS composed of genome-wide significant independent variants explained limited phenotypic variation (partial R² range: 0.008-1.6%), PRS were constructed restricting to HapMap3 variants plus all variants GWAS with P-values<0.05, resulting in a total of 1,457,694, 1,499,770, 1,538,890 variants for isoleucine, leucine, and valine include, respectively.” (line 382-384).

Second, the conditional F-statistic should include the covariance matrix between the exposures, when the exposures come from the same data source, as here. Please clarify whether any such covariance matrix was used and if not please explain the implications for the multivariable F-statistics.

As described by Sanderson et. al., the covariance matrix is required for the Cochran’s Q-statistic to test for instrument strength and validity when using *two-sample summary data*. Sanderson et. al specify alternative tests, specifically the Sanderson-Windmeijer F-statistic, which does not require the covariance matrix in the case of single-sample MVMR. Because we are applying single-sample MVMR, we are using the Sanderson-Windmeijer F-statistic, negating inclusion of the covariance matrix between the effect of the instrumental variables on each exposure (lines 226-232, 323-327). Indeed, because knowledge of the covariance matrix is a “limitation of the new tests we develop for two-sample summary data”, we were able to circumvent this limitation by using single-sample MVMR.

Third, please ensure that the number of variants for each exposure is included in the multivariable analysis given in the results section of the main text.

Please see lines 383-387.

2. Phewas studies are intended to be comprehensive, so the restriction to a relatively small number of curated phenotypes does not seem consistent with conducting a Phewas. It would make more sense to explicitly restrict the Phewas to items that could be the result of BCAAs. It would also be very helpful to give a definition of ‘curated’ and then explain how it corresponds to other criteria for conducting a valid mendelian randomization (MR) study, such as avoiding covariable adjustment, and for conducting valid studies of causal inference, such as counting exposure time correctly. For example, reports of genetically predicted reported medication use generate a lifelong exposure when medication use is not usually life long. It would be very helpful to give results for a comprehensive set of phenotypes, selected on criteria concerning the relevance and validity of each available item. It would also be helpful to validate the findings using other data sources for the outcomes.

We have revised our manuscript to more clearly describe the goals of our manuscript, e.g., to identify inconsistencies in the observational literature examining BCAAs, not to comprehensively catalog BCAA phenotypic effects. Given the very high significance of our results and consistency of our main findings –

that BCAAs harbor distinct and often directionally inconsistent effects - across different phenotype classes and that these observations are consistent with the experimental literature, we have not sought replication.

3. The paper aims to present the direct effects of three BCAAs. A direct effect is the effect of exposure on outcome absent the mediator, i.e., after accounting for a mediator. This means that the authors are postulating that the effect of each BCAA could be mediated by the other two BCAAs, which is not possible, as explained previously. Moreover, the new Figure 3 showing the analytic structure does not show any mediation, instead it looks like the authors are thinking that the three BCAAs do not cause each other but may be subject to genetic pleiotropy. Specifically, the new Figure corresponds to Figure 4 scenario 3 of this paper laying out the questions addressed by multivariable MR. The paper needs to be rewritten either as assessing the independent effect of each BCAA, assuming genetic pleiotropy only along with physiological justification for this approach, or some other justifiable and coherent underlying structure linking the three BCAAs needs to be specified and tested. Logically, it cannot be the current presentation where it appears that each BCAA mediates the other two. All relevant language throughout the text needs to be amended to reflect the question addressed

We agree with the reviewer that there are numerous different pathways through which the BCAAs could impact outcomes of interest. While in our new Figure 3, we do show an example which aligns with Scenario 3 shown in Figure 4 in the Sanderson paper, we do anticipate that there could be other hypothesized causal pathways. We are limited by the observational nature of this study, and unable to determine the exact pathway and even if we could do so, it is not feasible to illustrate each pathway given there likely could be some heterogeneity in terms of the exact pathway across a number of outcomes in this paper.

We have added additional language to the discussion highlighting this point as a weakness. However, as Table 1 in the Sanderson paper demonstrates, MVMR always provides an unbiased estimate of the direct effect of an exposure, in their case X_1 , on the outcome when examining simulations generated under the four scenarios shown in Figure 4. In several cases, including Scenario 3, Sanderson *et al* illustrate that this does estimate the direct and total effects for exposure. Therefore, while in many cases, the MVMR does estimate the total and direct effect, in the specific cases where it provides only the direct effect it still does provide important insight that cannot be assessed in an unbiased way, as shown in Sanderson's Table 1, via univariate MR. We have however, added some language to the discussion to highlight that more research needs to be done, likely individually for each health outcome to better understand upstream pathways (see lines 222-226).

4. This is a one-sample MR study, which at the limit equates to a two-sample study. However, biases are possible in smaller studies, which should be discussed here.

We agree there are biases in smaller studies and have added language as such to the discussion. However, we also wanted to highlight the benefits to one-stage over the two-stage methods, which include smaller variances of the causal parameters of interest and avoidance of the assumptions of similarity of populations required when using non-overlapping populations in two sample calculations (lines 228-232).

5. The explanation for using prevalent cases, i.e., that longitudinal data is often unavailable for the entire cohort, is confusing. The UK Biobank participants were followed up through electronic record linkage to death records, cancer registries, primary care and hospital in-patient records <https://www.ukbiobank.ac.uk/explore-your-participation/following-your-health>. The UK has a national health service (NHS) with very limited provision of care outside the NHS. So, record linkage to care within the NHS and to death records should be fairly complete. Please can this point be clarified.

We only included prevalent cases for variables measured at study baseline, e.g., type 2 diabetes and chronic kidney disease. Incorporation of primary care data, for example, would be a massive undertaking that is outside the scope of this paper.

Reviewers' comments:

Reviewer #1 (Remarks to the Author):

The authors have adequately responded to the prior review comments.

Reviewer #2 (Remarks to the Author):

Thank you very much indeed, the methods and study design are now much clearer.

In particular, thank you for clarifying that the underlying causal model is provided in Figure 3. Figure 3 shows that the underlying model is that the BCAAs share genetic confounders but do not have causal relations with each other. In this context, there does not seem to be any mediation of the effect of one BCAA by another, because the BCAAs do not affect each other. As such, this study seems to be assessing total effects not direct effects. For clarity, please remove all reference to direct effects or specifically explain what direct effect has been assessed, how it relates to Figure 3, and why it is a direct effect, whenever direct effects are mentioned.

Maybe, I am wrong, but it looks like this study is examining causal effects, as given in the title. The new text at lines 49-52 says that the high correlation between BCAAs creates bias in estimation. Correlation is not a common cause of exposure and outcome, so does not necessarily create bias. Shared causes of BCAAs would create bias. Please clarify the text throughout to explain that the issue is highly confounded data not highly correlated data or explain why non-causal correlations bias estimation of causal effects.

The question about the criteria for selection of the phenotypes and the validity of some phenotypes remains unanswered. Lines 290 tells us that phenotypes were selected, and lines 291 tells us that "nutrition/dietary data, blob/bulk and elements measured in fewer than 5000" were excluded. This information does not explain the rationale for phenotype selection. Line 302 gives a clue when it says "to increase capture of clinical phenotypes". However, the rationale for choosing the phenotypes should be explained at the start and then the criteria for selecting phenotypes explained. In addition, invalid clinical phenotypes should be removed, such as medication use as explained before. Please also reflect on the strength and weaknesses of the choice of phenotypes in the discussion.

Please clarify and justify consideration of multiple comparisons or otherwise.

The new limitation at lines 223-226 should be deleted, as it seems inconsistent with the design.

Finally, the cutoff (p-value 0.05) for selecting genetic variants for the BCAAs is quite unusual. So, it would be very helpful to include a sensitivity analysis with a different means of selecting instruments for each BCAA, or a different p-value cut-off.

Reviewers' comments:

Reviewer #1 (Remarks to the Author):

The authors have adequately responded to the prior review comments.

Thank you for taking the time to review our manuscript multiple times and providing valuable feedback.

Reviewer #2 (Remarks to the Author):

Thank you very much indeed, the methods and study design are now much clearer.

In particular, thank you for clarifying that the underlying causal model is provided in Figure 3. Figure 3 shows that the underlying model is that the BCAAs share genetic confounders but do not have causal relations with each other. In this context, there does not seem to be any mediation of the effect of one BCAA by another, because the BCAAs do not affect each other. As such, this study seems to be assessing total effects not direct effects. For clarity, please remove all reference to direct effects or specifically explain what direct effect has been assessed, how it relates to Figure 3, and why it is a direct effect, whenever direct effects are mentioned.

As requested by the reviewer, we now describe the effects as total effects. We have removed all reference to direct effects and replaced this with total effects. Additionally, we have added at lines 304-306, additional clarification stating, "In scenarios such as those specified in Figure 3, MVMR can be used to estimate the total causal effects of each BCAA."

Maybe, I am wrong, but it looks like this study is examining causal effects, as given in the title. The new text at lines 49-52 says that the high correlation between BCAAs creates bias in estimation. Correlation is not a common cause of exposure and outcome, so does not necessarily create bias. Shared causes of BCAAs would create bias. Please clarify the text throughout to explain that the issue is highly confounded data not highly correlated data or explain why non-causal correlations bias estimation of causal effects.

We have made numerous changes to the abstract, introduction, results, discussion, and methods to provide the clarification requested. Because these changes are extensive, we point the reviewer to the marked copy rather than specific line numbers.

The question about the criteria for selection of the phenotypes and the validity of some phenotypes remains unanswered. Lines 290 tells us that phenotypes were selected, and lines 291 tells us that "nutrition/dietary data, blob/bulk and elements measured in fewer than 5000" were excluded. This information does not explain the rationale for phenotype

selection. Line 302 gives a clue when it says “to increase capture of clinical phenotypes”. However, the rationale for choosing the phenotypes should be explained at the start and then the criteria for selecting phenotypes explained. In addition, invalid clinical phenotypes should be removed, such as medication use as explained before. Please also reflect on the strength and weaknesses of the choice of phenotypes in the discussion.

Reflecting on this comment, we believe our use of the terms “phewas” or “phenome-wide” introduced misunderstanding. Agreeing with the reviewer, and because we do not make an exhaustive effort to catalog all phenotypes available in the UK Biobank, we have removed all references to “phewas” or “phenome”. We also have updated our rationale to phenotypic selection (lines 262-286).

Please note that we have retained the medication phenotypes to enable capture of disorders like hypothyroidism and gastro-esophageal reflux disease (Yu, 2019).

Wu Y, Byrne EM, Zheng Z, Kemper KE, Yengo L, Mallett AJ, Yang J, Visscher PM, Wray NR. Genome-wide association study of medication-use and associated disease in the UK Biobank. *Nat Commun.* 2019 Apr 23;10(1):1891. doi: 10.1038/s41467-019-09572-5. PMID: 31015401; PMCID: PMC6478889.

Please clarify and justify consideration of multiple comparisons or otherwise.

Given the goal of this paper was concept rather than hypothesis driven (i.e., examine whether discrepancies between experimental and observational BCAA studies reflected biasing effects of shared genetic and lifestyle factors), statistical significance of BCAA direct causal effects was determined using a conservative Bonferroni correction accounting for 441 phenotypes and three BCAAs (i.e., $0.05/[441*3] = 3.78 \times 10^{-5}$).

The new limitation at lines 223-226 should be deleted, as it seems inconsistent with the design.

We deleted this limitation, as requested.

Finally, the cutoff (p-value 0.05) for selecting genetic variants for the BCAAs is quite unusual. So, it would be very helpful to include a sensitivity analysis with a different means of selecting instruments for each BCAA, or a different p-value cut-off.

Please see newly added clarifications (lines 366-383). Briefly, as a first step, we used a genome-wide p-value threshold of $p < 0.05$, in combination with HapMap3 variants, to narrow down genetic variants as potential candidates for inclusion as instrumental variables. In this step, we used a p-value cutoff of 0.05 in order to maximize predictive ability, particularly given the high correlation between BCAAs. The actual process of selecting genetic variants, specifically the generation of a PRS score for each BCAA, was done using the Crosspred method (Mak 2018). This method implements lasso regression, which utilizes a variable selection framework, and generates a PRS that outperforms p-value thresholding methods in terms of prediction.

Additional benefits of this approach include the use of cross-validation to address overfitting. Please note that selection of method was guided by final recommendations from Burgess et. al 2011, which states the best estimates of causal assumption would be generated by selecting “only the most important IVs, based on biological knowledge and external information,” especially given we did not have an independent discovery dataset available.

Burgess, Stephen, Simon G. Thompson, and Crp Chd Genetics Collaboration. "Avoiding bias from weak instruments in Mendelian randomization studies." *International Journal of Epidemiology* 40.3 (2011): 755-764.

Mak, Timothy Shin Heng, Robert Milan Porsch, Shing Wan Choi, and Pak Chung Sham. "Polygenic scores for UK Biobank scale data." *BioRxiv* (2018): 252270.

REVIEWERS' COMMENTS:

Reviewer #2:

None